# Extremophiles in Soil Communities of Former Copper Mining Sites of the East Harz Region (Germany) Reflected by Re-Analyzed 16S rRNA Data

**DOI:** 10.3390/microorganisms9071422

**Published:** 2021-06-30

**Authors:** J. Michael Köhler, Nancy Beetz, Peter Mike Günther, Frances Möller, Jialan Cao

**Affiliations:** Department of Physical Chemistry and Microreaction Technology, Institute for Micro and Nanotechnologies/Institute for Chemistry and Biotechnology, Technical University Ilmenau, 98693 Ilmenau, Germany; nancy.beetz@tu-ilmenau.de (N.B.); mike.guenther@tu-ilmenau.de (P.M.G.); frances.moeller@tu-ilmenau.de (F.M.); jialan.cao@tu-ilmenau.de (J.C.)

**Keywords:** soil bacteria, ancient mining places, halophiles, acidophiles, thermophiles, psychrophiles, NGS

## Abstract

The east and southeast rim of Harz mountains (Germany) are marked by a high density of former copper mining places dating back from the late 20th century to the middle age. A set of 18 soil samples from pre- and early industrial mining places and one sample from an industrial mine dump have been selected for investigation by 16S rRNA and compared with six samples from non-mining areas. Although most of the soil samples from the old mines show pH values around 7, RNA profiling reflects many operational taxonomical units (OTUs) belonging to acidophilic genera. For some of these OTUs, similarities were found with their abundances in the comparative samples, while others show significant differences. In addition to pH-dependent bacteria, thermophilic, psychrophilic, and halophilic types were observed. Among these OTUs, several DNA sequences are related to bacteria which are reported to show the ability to metabolize special substrates. Some OTUs absent in comparative samples from limestone substrates, among them *Thaumarchaeota* were present in the soil group from ancient mines with pH > 7. In contrast, acidophilic types have been found in a sample from a copper slag deposit, e.g., the polymer degrading bacterium *Granulicella* and *Acidicaldus*, which is thermophilic, too. Soil samples of the group of pre-industrial mines supplied some less abundant, interesting OTUs as the polymer-degrading *Povalibacter* and the halophilic *Lewinella* and *Halobacteriovorax*. A particularly high number of bacteria (OTUs) which had not been detected in other samples were found at an industrial copper mine dump, among them many halophilic and psychrophilic types. In summary, the results show that soil samples from the ancient copper mining places contain soil bacterial communities that could be a promising source in the search for microorganisms with valuable metabolic capabilities.

## 1. Introduction

Microbial diversity is very important for a healthy environment and for fertile soils [1,2]. Besides ubiquitous bacteria, less abundant types also contribute to the diversity in soil bacterial communities [3,4]. Among them, extremophilic bacteria can also be found. It is well known that the acid drainage of active mines causes the development of acidophilic bacterial communities, which are of interest for bioleaching [5,6]. However, many microbial types related to lower pH are also found in many other and in non-acidic soils. Many families and genera of Acidobacteria, for example, are very abundant and interesting for usable bioproducts, but the investigation of their physiology and ecology is hindered by difficulties in cultivation [7,8].

The special ecological situation and the prevalence of acidophilic bacteria in industrial mining areas can easily be explained by the acidification due to bioleaching and biooxidation of minerals. In contrast, ancient mining areas usually show a special pattern of vegetation and soil development, although former mining activities ceased centuries ago [9]. Obviously, their soils and soil microbial communities represent a typical example of ecological memory. Traces of early human activities, which changed the structure and character of soils, can be stored over a long time and are reflected in the local compositions of the soil bacterial communities. This “ecological memory of soil” was observed, for example, on prehistorical settlements, working and burial places [10,11,12,13]. The understanding of “man-made environments”, mining-related acidic soils among them, for example, remains an urgent challenge for investigations of soil microbial communities up to now [14].

An ecological memory effect could also be observed on ancient mining sites of the East Harz copper mining region (Sachsen-Anhalt, Germany) [15,16]. The metal mining in this region dates back to the Middle Ages but continued up to the 1990s [9]. As expected, the abundance of different bacterial types is strongly dependent on pH, which was confirmed by comparing some of these East Harz mining places with other soil samples from Central Germany. Besides acidophilic and halophilic types, the ancient mining places also yielded bacterial DNA of thermophilic taxonomical units. During an investigation of soil bacterial communities of a group of mining and smelting places, special types of less-abundant extremophiles were found, among them *Hadesarchaeota* [15].

The abundance of special bacterial types, and of extremophiles in particular, is not only of interest for the biodiversity and robustness of the local ecological situation [17], the remediation of industrial used areas, and their sustainable usage. It is also very important for the search of microorganisms with special metabolic activities which could be applied in new biotechnological processes [18]. Therefore, here, 16S rRNA profiling data of soil samples from East Harz copper mining places are analyzed and partially re-analyzed for their possible relevance in searching for further interesting extremophiles. The soil samples were collected, originally, in searching for heavy metal-tolerant bacterial strains, but then found to be of interest with respect to halophilic, acidophilic, and thermophilic bacteria too [15].

## 2. Materials and Methods

### 2.1. Soil Samples

For the investigation, 16S rRNA data of 18 soil samples from pre-industrial and early industrial copper mining places in the East Harz region have been selected. Six samples originate from deposits surrounding pre-industrial mine shafts (Wolferode, Wimmelburg, Wiederstedt, Hergisdorf), six samples are from deposits near early industrial mining shafts (Hettstedt and Welfesholz) and six samples are from deposits surrounding earlier mine shafts (Pölsfeld, Rodishain, and Uftrungen, approximately 15th/16th century). The sample material (No. 1–19) is not any kind of typical natural soil, but surface material from the mine dumps, was brought from the ground to the surface during the site-related mining phases. The soil material was taken during dry weather periods. For sampling vegetation-free spots of the surface have been used. The material was stored in sterile 50-mL sample tubes. The samples were dried at room temperature. The sampling locations, their GPS coordinates, pH values, and electrical conductivity values are given in Table 1. In addition, a map showing the sampling sites is provided in the Appendix A.

The idea behind the selection of soil samples was to search for special bacterial types related to mining activities in the past and compare them with some samples from non-mining places. Thus, small mining places, in particular from the early mining activities in the 15th–18th century (Uftrungen, Rodishain, Pölsfeld, Hergisdorf, Wolferode and Wiederstedt), early industrial mining sites (1st half of 19th century; Burgörner, Hettstedt, Welfesholz) and two industrial places (20th century) were chosen.

In addition to the soil samples from ancient mining areas, one special soil sample was included which had been collected at the deposit pile base of the industrial copper mine of Nienstedt near Sangerhausen (Southeast Harz region, No. 19). Beside this sample, six other samples are included as comparative samples (control samples) which originate from forest-covered areas of ancient ramparts: three from limestone substrates (No. 20–22) and three from acid mineral substrates (quartzite and sandstone, No. 23–25). The comparative samples were chosen from a sample collection of different parts of Thuringia in order to have different regions and different geological situations. The samples from line stone substrates (pH > 7.5) came from northwest Thuringia (Haynrode), North Thuringia (Burgwenden) and from the Saale valley (Kahla). The samples from acid substrate soils were taken from southwest Thuringia (Northern Rhön, Völkershausen, basalt), South Thurigina (Suhl, sandstone), and Northeast Thurigina (Rastenberg, sandstone).

### 2.2. DNA Extraction and Sequencing

For DNA isolation, the Power Soil Isolation Kit (MO BIO, Carlsbad, CA, USA) was applied according to the protocol of supplier. The polymerase chain reactions (PCR) were carried out using an Edvocycler (Edvotek, Washington, DC, USA). Gel electrophoresis in 1.2% agarose gels was used for checking the success of each PCR amplification step. AMPure XP Beads (Beckman Coulter, Brea, CA, USA) were applied for the purification of the primary PCR products as well as for the completed pooled libraries according to the manufacturer’s protocol.

The required adaptor primers for Amplicon PCR A519F-Ad (5′ TCGTCGGCAGCGTCAGATGTGTATAAGAGACAGCAGCMGCCGCGGTAA 3′) and Bact_805R-Ad (5′-GTCTCGTGGGCTCGGAGATGTGTATAAGAGACAGGACTACHVGGGTATCTAATC 3′) were supplied by Eurofins Genomics (Ebersberg, Germany). A concentration of 100 pmol/µL was applied. The reaction mixtures (total volume of 50 µL per reaction) were composed by following components: 1 µL of DNA isolation eluate, 2 mM MgCl_2_, 200 µM PCR nucleotide mix, 1.25 Units GoTaq G2 Flexi DNA Polymerase, nuclease-free water (all reagents from Promega, Madison, WI, USA), and 1 pmol/µL of each primer. For PCR amplification, following program settings were used: start denaturation: 5 min at 94 °C; 30 amplification cycles involving 30 s denaturation at 94 °C, 30 s annealing at 50 °C and 30 s primer extension at 72 °C. The temperature cycling was finished with a final extension at 72 °C for 5 min.

The required eight forward indexing primers and twelve reverse indexing primers for index PCR were delivered by integrated DNA technologies (Coralville, CA, USA). The primers were applied in a concentration of 1.25 pmol/µL. The following composition was used for the index PCR: total volume of 25 µL per reaction; 2.5 µL of Amplicon PCR product, 2.5 mM MgCl_2_, 300 µM PCR nucleotide mix, 0.5 Units GoTaq G2 Flexi DNA Polymerase, nuclease-free water (all reagents from Promega, Madison, WI, USA), and 125 nmol/L of each of the two primers of the respective indexing primer combination.

For index primer PCR, following program settings were applied: start denaturation: 3 min at 95 °C; 30 amplification cycles involving 30 s denaturation at 95 °C, 30 s annealing at 55 °C, and 30 s primer extension at 72 °C. The temperature cycling was finished with a final extension at 72 °C for 5 min.

The sequencing was done by GATC Biotech (Konstanz, Germany) and Eurofins (Ebersberg, Germany) by next generation sequencing on an Illumina HiSeq.

### 2.3. Processing of NGS Data

The software tool “Galaxy” [19] was applied for checking the quality of the NGS sequence data. All investigated data sets are characterized by a high median quality score.

The data processing was based on the automatic software pipeline of the SILVAngs data analysis service. It allowed a detailed community analysis sequencing data [20,21]. All data files from the NGS analyses were first converted from fastq file format to fasta file format by use of the fastq-to-fasta converter tool “phred33 conversion” (MR DNA Lab). The reads are aligned by SILVA Incremental Aligner (SINA SINA v1.2.10 for ARB SVN (revision 21008)) against the SILVA SSU rRNA SEED and quality controlled (applying minimal identity criteria of 98%). Reads with a low alignment quality, less than 50 aligned nucleotides and reads with more than 2% of ambiguities, or 2% of homopolymers, respectively, were not processed. For classification, a BLAST search with the database SILVA Ref (release 132) was applied. For all analyses, the preset parameter sets of the settings page and with the SILVAngs database release version 132 were applied [21].

In many cases, the data from sequencing allow an assignment of bacterial groups down to the genus level. In some other cases, it is only possible to identify higher taxonomic levels. The lowest level identified is referenced as “operational taxonomic unit” (OTU). The comparison of soil bacteria and the compositions of microbial communities is based on the genera as far as possible or on OTUs. PCA was performed by using the Matlab standard procedure.

## 3. Results and Discussion

### 3.1. Phyla and Dominant Taxonomical Units

The phyla Proteobacteria, Planctomycetes, Bacteroidetes, Chloroflexi and Actinobacteria are abundant in all investigated samples. Besides, Verrucomicrobia, Gemmatimonadetes, Acidobacteria, and Archaea are abundant in most samples. However, the ratios of phyla vary significantly between the different samples (Figure 1a). The comparative samples from limestone substrates (No. 20–22) show significantly higher contents of Bacteroidetes than the comparative samples from the acidic soil substrates (No. 23–25). In contrast, the samples 23 and 24 (pH 4.0) show an enhanced content of Acidobacteria.

The most marked differences in comparison with all other samples are observed for sample No. 19 (from the industrial copper mine deposit). It is characterised by a strong dominance of Bacteroidetes and very low contents of Acidobacteria and Archaea. In difference to the other samples, No. 3 is marked by a particularly high dominance of Chloroflexi.

A clear differentiation of samples is possible by the abundance at a lower taxonomical level (Figure 1b). Particularly high contents in the group of four frequent OTUs (Sphingomonas, IMCC26256, Ralstonia and Ilumatobacteraceaea) are observed in the samples No. 5, 7, 8, and 10–12 (belonging mainly to sample group C), which differ from all other samples by these types (Figure 1b). This difference between samples seems to be related to some extent to the pH of the soil, because the samples No. 7–12 (all group C) showed the highest pH values (see Table 1). However, it should be noted that the samples No. 9 is not included in this group, despite its pH is also above 8. The abundances of Sphingomonas in some samples could be of special interest because their metabolic potential is important for the whole community, for example by production of phytohormones as gibberellines and indol acetic acid [22], which might also be interesting for the future improvement of soil fertility and remediation by means of biotechnological products. The bacterial communities of some of the other samples are dominated by one or a few abundant OTUs (see Appendix A). For example, sample 16 supplied in 26% of the reads the OTU Candidatus Ud. (Chthoniobacteraceae) and sample 3 supplied in 36% of reads two OTUs of Ktedonobacteraceae. From this table, it can be seen clearly, that the samples No. 5, 7, 8, 10, 11, 12 (mostly group C) are distinguished from the other samples, which are closer related to the comparative samples from limestone substrates (Group F). Appendix A also clearly shows the special situation of samples 3 and 19.

A comparison between the sample groups (A–G) was based on a logarithmic abundance parameter *r* defined from the ratio of reads per OTU (*N_i_*) to the total of reads *N_tot_* per sample:(1)r=log(1+1,000,000∗Ni/Ntot)

Principle component analyses (PCA) of sets of most frequent OTUs using this parameter *r* reflect significant differences between the sample groups. The strength of distinguishing these groups depends on the set of included OTUs (Figure 2). The samples No. 3 (industrial slag deposit) and 9 (industrial mine dump) are clearly separated from the other samples if the 100 most abundant OTUs are considered (Figure 2a). No. 3 is also clearly separated from the others if the 20 most abundant OTUs are regarded (Figure 2b). Group B (pre-industrial mines, pH > 7.5) shows close relation to the comparative sample group F (limestone substrates) as could be expected. The comparative sample group from acid substrates (G) is always well separated from the other groups.

A distinguishing of the groups (A–G) is also possible by using sets of selected OTUs. The PCA including abundances of 17 OTUs shows two separate clusters of group F and four samples of group C. The other communities are found in a larger third cluster, but with a certain separation between Groups A, B, F, but including two samples of group C, too (Figure 3a). A better separation of sample group A from the other groups succeeded by correlation of r-values for selected OTUs, e.g., *Kineosporiae*, uncult., and *Parafilimonas* (Figure 3b). Group 3 can be distinguished from groups A, F, and G in the correlation plot of *Ralstonia* and *Acidobacteria*, Subgroup 2 (Figure 4a). Groups A and B are also well distinguishable by the correlation plot of *Hyphomonadaceae SWB02* and *Terrimonas* (Figure 4b).

### 3.2. Abundances of Taxonomical Groups Related to Extremophilic Characters

#### 3.2.1. Acidobacteria, Acidimicrobiia and Other Actinobacteria

The bacteria of phylum *Acidobacteria* and the class *Acidimicrobiia* are of interest because many acidophilic species are found in these groups. In our case, the abundances of bacteria of these groups over the samples from the mining areas and the comparative samples do not reflect a clear relation to sample pH (Figure 5). Beyond this general point of view, considerable differences between the samples are evident. The most remarkable is the very low *Acidobacteria* content in sample 19.

Clearer differences as in the absolute number of reads for *Acidobacteria* in relation to the origin of soil samples were observed for the abundance of different orders and related lower taxonomic levels of this phylum. This distribution is shown for a selection of 10 of the investigated samples (Appendix A): The comparative samples from acid soil (No. 24 and 25) are dominated by *Acidobacteriales* in combination with a high content of *Solibacterales* and very low contents of *Blastocatellales* and *Acidobacteria Subgroup 6*, which are abundant in most of the other samples. The large part of other types in these samples is mainly related to *Acidobacteria Subgroup2*. A considerably high proportion of this group was also found in sample 19 (industrial mine dump, Nienstedt). The other samples (without sample 3) are marked by a common dominance of *Blastocatellales* and *Subgroup 6* with *Acidobacteriales*. This picture matches the slightly higher pH of these samples.

The general abundances of *Actinobacteria* and *Chloroflexi* also varied between the samples, but without significant correlation with soil pH or conductivity (Figure 6). However, in analogy to the *Acidobacteria*, remarkable differences could be found at a lower taxonomic level (Appendix A). Thus, sample 19 is very strongly dominated by a group of uncultivated *Actinomarinales*. In contrast to all other samples, the group IMCC26256 is less abundant in the bacterial community of this sample. The last-mentioned group, together with an uncultivated group of *Acidimicrobia*, dominates the comparative samples from acid soil. These samples show very low contents of uncultivated *Ilumatobacteraceae* as well as uncultivated *Microtrichales*, which is mostly in contrast to the other samples. There are some additional striking differences in the abundances of different types of *Acidimicrobiia*, among them the dominance of *Iamia* in sample 3 and the dominance of *Ilumatobacteraceae* in sample 8 and 10. While *Iamia* was isolated from sea cucumber [23], *Ilumatobacter* was found in estuary sediments [24], which is evidence of salt tolerance.

Besides *Acidimicrobiia*, the other classes of Actinobacteria reflect some typical differences, too (Appendix A). The largest proportions of *Corynebacteriales* were found in sample 3 (pH 5.57) and in the communities of comparative samples from acid soils (No. 24 and 25). Sample 3 and 19 are noted by a high content of *Micrococcales*. Both types are abundant in wet soils, in general, and are marked by a high GC content in the DNA.

#### 3.2.2. Abundances of Chloroflexi

Despite the differences in the total abundance of *Chloroflexi*, the examples of soil samples from the ancient mining places as well as from the industrial mine dump (Appendix A) reflect a high abundance of *Ktedonobacteria*. The samples 8, 10, and 11 (pH values above 8) show significant proportions of the groups Gitt-GS-136 and KD4-96, which are also abundant in the comparative samples from limestone substrate, but less abundant in the samples 3, 16, and 19 as well as 24 and 25 (acid soil samples). The high content of “others” in the comparative samples from the acidic soil (Appendix A) is mainly due to the group Chloroflexi AD3, which is less or not present in the other samples. Sample 3 only showed 100 reads of this group. This type was found earlier in acid mine drainage and represents obviously acidophilic strains of soil bacteria [25].

#### 3.2.3. Special pH-Dependent OTUs

There exist a number of OTUs which are moderately present in the majority of the samples from ancient mining areas (without sample 3) and in all comparative samples from limestone areas (No. 20–22), but completely absent in the acid comparative samples (No. 23–25) This characteristic non-acidophilic behaviour was found in the investigated bacterial communities, e.g., for *Longimicrobiaceae*, for *Chloroflexi* group P2-11E, for the *Planctomycetes* group OM190, for the *Sphingobacteriales* KD3-93, for *Kribella,* and for the *Chloroflexi* group OLB14, with slightly reduced abundance for *Lysobacter* and the groups SM1A02 and NS11-12 marine groups (Appendix A).

The opposite situation is observed with a group involving *Acidipilia*, *Edaphobacter*, *Granulicella*, an acidophilic polymer degrading bacterium, *Actinospica*, *Acidicaldus*, and *Roseiarcus*, which are more abundant in the communities from the acidic comparative samples and partially in sample 3 (Appendix A). Some further OTUs (AAP99, JG36-TzT-191, *Thaumarchaeota* 1.1c) are absent in the comparative samples from limestone soil (No. 20–22) but appear on some ancient mining areas which show a soil pH between 6 and 7. *Edaphobacter* was isolated from forest soil and described as adapted to a neutral to slightly acidic environmental situation. Five strains of *Granulicella* were found to be adapted for acidic conditions (pH optimum at 3.8–4.0). In addition, they are psychrotolerant (2–33 °C) and show hydrolysis capability for pectin, xylan, laminarin, lichenan, and starch [26]. *Actinospica* was isolated from forest soil and characterized as an acidophilic bacterium [27]. *Acidicaldus* was isolated acidophilic (growing fast between pH 2.5 and 3) and thermophilic (optimum between 50 and 55 °C [28]. *Roseiarcus* was isolated from a methanotrophic microorganism consortium and showed a slightly acidophilic characteristic [29].

#### 3.2.4. Halo- and Psychrophilic Bacterial Soil Community in a Sample from Industrial Mine Dump Nienstedt

Sample No. 19 has a special position between the soil samples of the ancient mining areas (No. 1–18) and the comparative samples (20–25). In contrast to the ancient shaft hole places and small mine dumps, the base area of the industrial mine dump has no vegetation and nearly no humus formation. The sample is marked by a moderate pH (7.66) but has a very high electrical conductivity compared to all other samples, indicating a high concentration of mobile ions. This situation is clearly reflected by strong differences in the composition of the soil bacterial community. In sample 19, many OTUs were found with mediate or high read numbers, which are very rare or completely absent in all other samples (see Appendix A). The special soil communities of this sampling site could be explained by the special situation of this place. On the one hand, this large mine dump was operated in the 20th century and has been closed for some decades. This is in strong contrast to the small mines and dumps from the early industrial and the pre-industrial era, which have not operated in several centuries. The place (No. 19) has nearly no vegetation and therefore it was surprising to see the large spectrum of different OTUs in the NGS data. In addition, this place showed the highest electrical conductivity of soil speaking for a high content of soluble salts. This fact explains the high content of halophilic types, but in addition, evinces another character of the microbial community as whole.

It is less surprising that many salt-tolerant or halophilic bacteria are in the above-mentioned group of microorganisms. This group involves mostly bacteria firstly described from sea water or marine environments, in general, among them the *genera Salinisphaera* [30], *Marinimicrobium* [31], *Marinoscillum* [32], *Roseivirga* [33], *Arenibacter* [34], *Cellulophaga* [35], *Confluentibacter* [36], *Gaetbulibacter* [37], *Maribacter* [38], Muricauda [39], *Tamlana* [40], *Balneola* [41], *Jannaschia* [42], and *Hoeflea* (formerly “*Agrobacterium*”) [43]. For *Muricauda*, it is noteworthy that it can use hexadecane as the sole carbon source [39].

More interestingly, the special group of bacteria from the industrial mine dump involves a considerable portion of psychrophilic bacteria firstly isolated from artic environments. These include the genera *Loktanella* [44], *Pricia* [45], *Cryomorpha* [46], and *Algibacter* [47]. In addition, there are several OTUs related to genera with pigment producing strains, for example, belonging to *Aequorivita* [48], *Salegentibacter* [49], *Subsaxibacter* [50], and *Arcticiflavibacter* [51]. It is important to mention that the majority of these found genera are represented in the investigated sample in a comparatively high number of reads. A noticeably high number of reads (626) was also found for *Psychroflexus*, which is famous for its special ability to produce eicosanopentaeonic acid, a polyunsaturated fatty acid [52]. Despite the occurrence of a lot of halophilic organisms in this sample, the content of psychrophilic genera confirms the special character of the related soil bacterial community. The coincidence of salt content, appearance of psychrophiles, and high probability of temporary exposure to heavy metals from the dump erosion support the formation of soil microbiomes of special compositions containing types, from which special metabolic features could be expected, e.g., related to metalloenzymes.

#### 3.2.5. Less Abundant Types with Special Tolerances and Metabolic Features

The NGS data reflect a set of less abundant further OTUs besides the taxonomic groups mentioned above, which are of interest due to their special tolerance features and promising physiological activities. An overview is presented in Appendix A. These OTUs are not “rare types” and most of them are also found in low read numbers in the group of comparative samples. Included are OTUs with these special properties described in the literature, which occur in at least 10 reads in total over all samples. Among them are thermophilic bacteria, such as *Tepidiphilus*, *Anaerolinea,* and *Alterococcus,* as well as salt-tolerant bacteria, such as *Fulvivirga*, *Tistlia*, *Lewinella*, *Marisediminicola*, *Salinispora*, and *Halobacteriovorax*. Several of these types have been reported to be able to degrade special substances, for example agar, gellan-gum, salicylate, molinate, alkanes, synthetic polymers like polyvinylalcohol and critical environmentally hazardous substances like aromatic and polycyclic hydrocarbons (Appendix A). The special types have different occurrence in the sample groups. Thus, the polyvinylalcohol-degrading *Povalibacter*, the halophilic bacteria *Lewinella* and *Halobacteriovorax* were found in group A (pre-industrial mines), whereas polymer-degrading acidophilic organisms among them *Thaumarchaeota* appeared in the soil of the slag deposit area (group D).

## 4. Conclusions

The analysis of the 16S rRNA data from the former copper mining places of the East Harz region reflects the occurrence of different groups of extremophilic bacteria, including acidophiles, halophiles, as well as thermophiles. Even the mining sites operated in the pre-industrial period (about 16th to 18th century) and the early industrial period (first half of 19th century) supplied bacterial types that differ in their occurrence or abundance from six selected comparative samples from non-mining sites. The occurrence of some taxonomic groups of acidophilic bacteria correlates with comparative samples from forest areas on acidophilic substrates, while others differ significantly from them. A considerably high portion of salt-tolerant, halophilic, as well as psychrophilic bacteria was exclusively found in a sample from the base of an industrial mine dump operated in the 20th century.

Among the identified genera, there are some that have been reported to have special metabolic features. This concerns the ability to metabolize various macromolecular carbohydrates as well as synthetic polymers, synthetic aromatic hydrocarbons, and polycyclic aromatic compounds. Some of the strains belonging to the genera found produce different types of pigments or other special metabolites, or a polyunsaturated fatty acid, for example. The differences in the composition of soil microbial communities and the abundances of genera known for extremophilic behavior are not related to soil pH exclusively and salt content. In addition, components of soil bacteria were found in the samples, indicating that different sampling sites can be valuable to the search for new strains with interesting metabolic properties.

It can be concluded that, besides the special situation in recently operated mines, mine dumps, and acid main draining, ancient mining sites with mining activities dating back over centuries could also be interesting places to find promising special soil bacterial communities, including extremophiles and bacterial strains with different tolerance features and interesting metabolic capabilities.

## Figures and Tables

**Figure 1 microorganisms-09-01422-f001:**
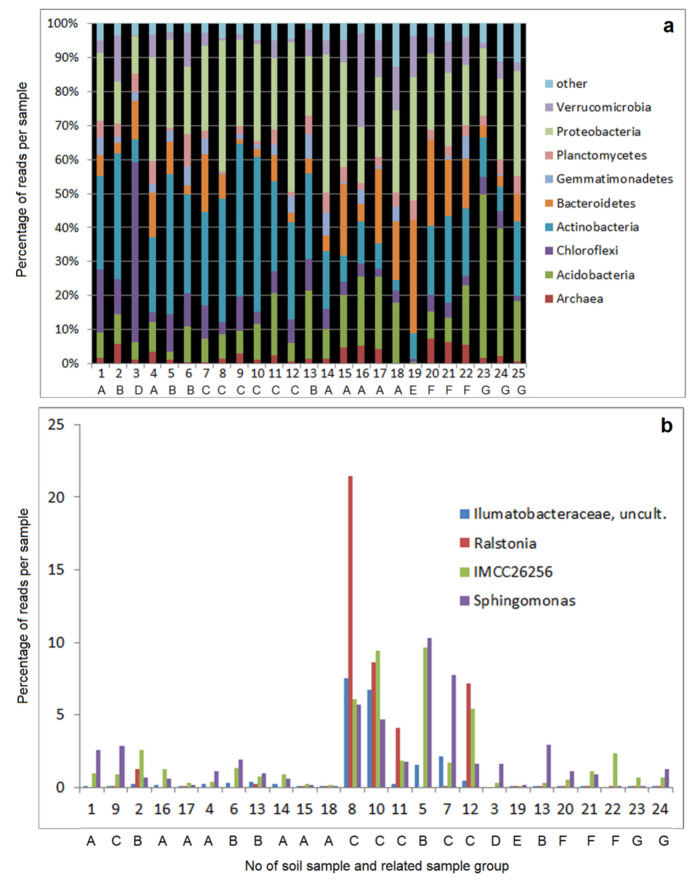
(**a**) Percentage of the most abundant phyla in the 19 investigated samples from formerly copper mining areas (No. 1–19, groups A–E) and the comparative samples (No. 20–25, groups F and G); (**b**) Domination of single OTUs: Percentages of the total number of reads for four single OTUs with high abundance in the samples 5 (Wiederstedt, group B, pH 7.72), 7 (Welfesholz, group C, pH 8.24), 8 (Welfesholz, group C, pH 8.78), 10 (Burgörner, group C, pH 8.85), 11 (Burgörner, group C, pH 8.28) and 12 (Hettstedt, group C, pH 8.01).

**Figure 2 microorganisms-09-01422-f002:**
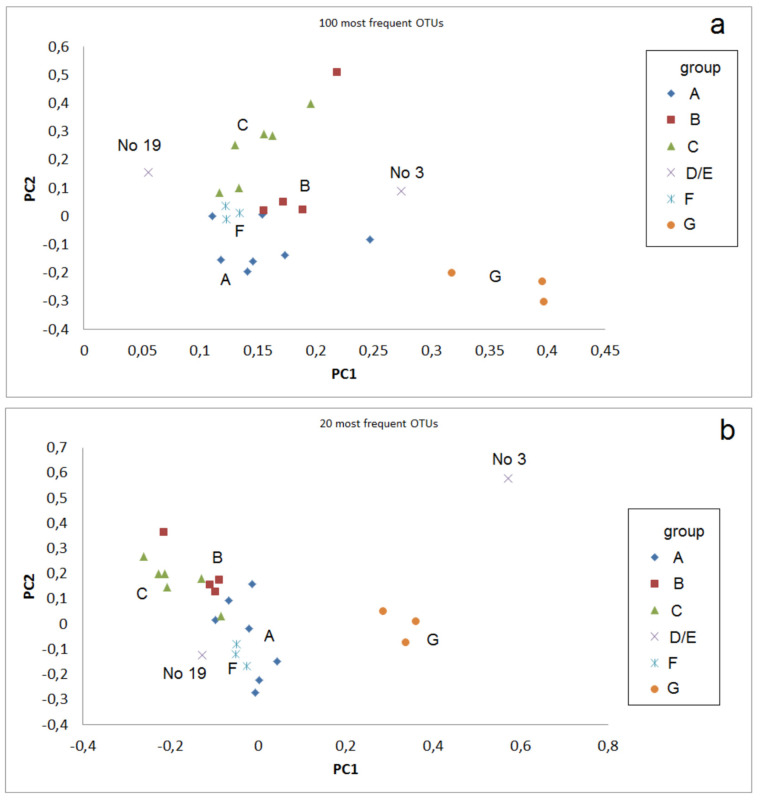
Principle component analysis of abundances in the sample groups by r-values (Equation (1)): (**a**) Principal component (PC) plot for PC1 and PC2 for the 100 most abundant OTUs; (**b**) Principal component plot for PC1 and PC2 for the 20 most abundant OTUs.

**Figure 3 microorganisms-09-01422-f003:**
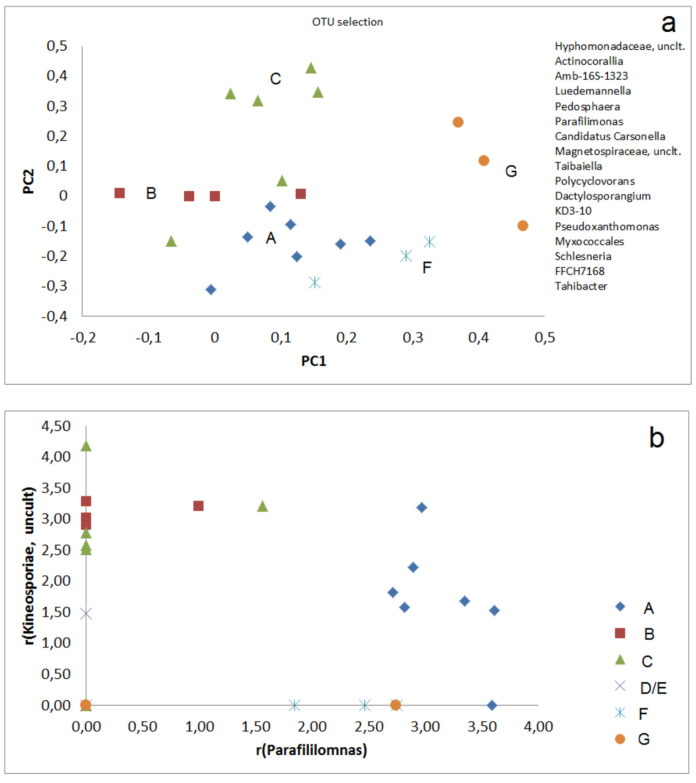
Distinguishing of sample groups by r-values (Equation (1)) of selected OTUs: (**a**) PCA of 17 selected OTUs; (**b**) Correlation of two selected OTUs (*Parafilimonas*, *Kineosporiae*).

**Figure 4 microorganisms-09-01422-f004:**
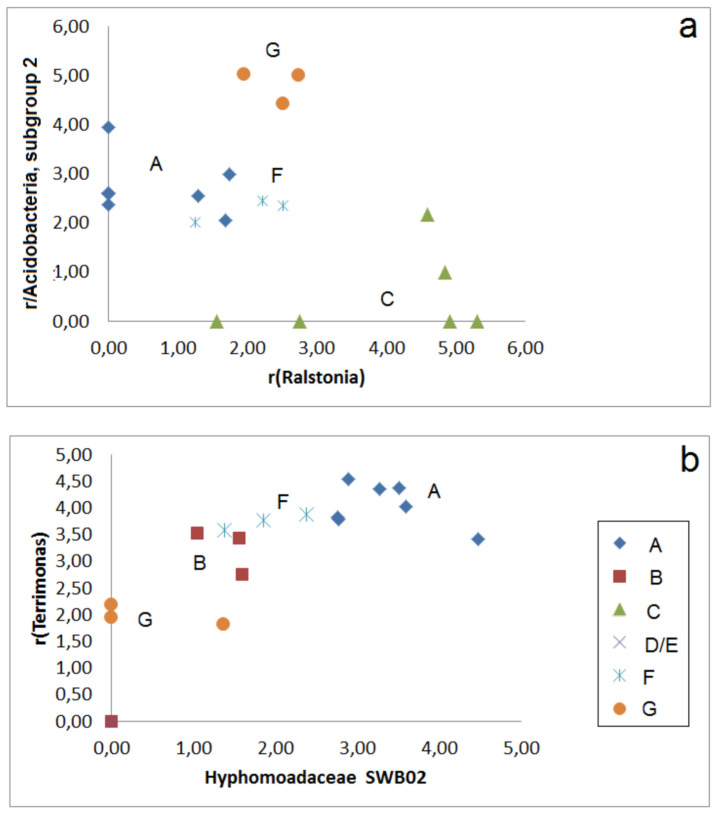
Correlation plots for abundance-related r-values for two pairs of selected OTUs: (**a**) *Ralstonia/Acidobacteria*, *subgroup 2*; (**b**) *Hyphomonadaceae SWB02*, *Terrimonas*.

**Figure 5 microorganisms-09-01422-f005:**
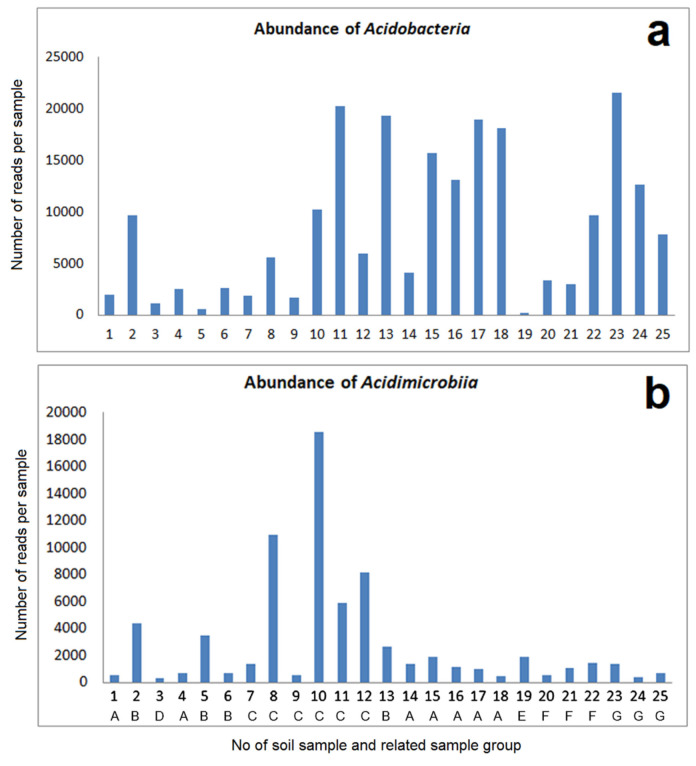
Highly abundant taxonomic groups (total number of reads per sample): (**a**) Acidobacteria; (**b**) Acidimicrobiia; The numbers on the x-axis are related to the sample numbers of Table 1.

**Figure 6 microorganisms-09-01422-f006:**
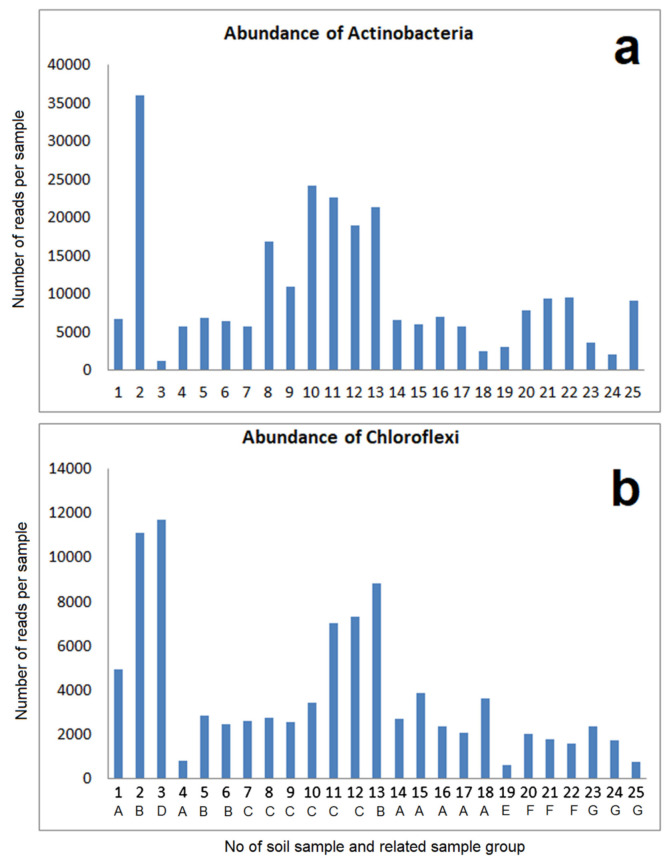
Highly abundant taxonomic groups (total number of reads per sample): (**a**) Actinobacteria; (**b**) Chloroflexi.

**Table 1 microorganisms-09-01422-t001:** Origin of samples (overview, for more details, please see Appendix A).

No.	Location	Group	pH Value	Electrical Conductivity
1	Wolferode, pre-industrial mine	A	7.15	225 µS/cm
2	Wolferode, pre-industrial mine	B	7.56	220
3	Wimmelburg, slag deposit	D	5.57	35.4
4	Hergisdorf, pre-industrial mine	A	7.11	724
5	Wiederstedt, pre-industrial mine	B	7.72	695
6	Wiederstedt, pre-industrial mine	B	7.63	201
7	Welfesholz, mine dump, early 19th century	C	8.24	94.5
8	Welfesholz, mine dump, early 19th century	C	8.78	69.1
9	Burgörner, mine dump, early 19th century	C	8.27	148
10	Burgörner, mine dump, early 19th century	C	8.85	82
11	Burgörner, mine dump, early 19th century	C	8.28	163
12	Burgörner, mine dump, early 19th century	C	8.01	237
13	Pölsfeld, pre-industrial mine dump	B	7.69	228
14	Rodishain, pre-industrial mine	A	6.98	331
15	Rodishain, pre-industrial mine	A	7.04	1240
16	Uftrungen, pre-industrial mine	A	6.23	229
17	Uftrungen, pre-industrial mine	A	6.79	367
18	Uftrungen, pre-industrial mine	A	7.34	344
19	Nienstedt, industrial mine dump	E	7.66	4677
**Comparative sites (limestone substrate, forest)**				
20	Burgwenden, Monraburg, prehist. rampart	F	7.84	183
21	Haynrode, Hasenburg, prehist. rampart, castle	F	7.82	246
22	Kahla, Dohlenstein, prehist. rampart	F	6.82	455
**Comparative sites (acid soil, forest)**				
23	Suhl, Lange Bahn	G	3.99	56.7
24	Völkershausen, Dietrich	G	4.01	231
25	Rastenberg, Streitholz	G	4.42	64.7

## Data Availability

The data presented in this study are available on request from the corresponding author.

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
