# Peer review of "Extremophiles in Soil Communities of Former Copper Mining Sites of the East Harz Region (Germany) Reflected by Re-Analyzed 16S rRNA Data"

_microorganisms, 2021, doi:10.3390/microorganisms9071422_

Round 1

Reviewer 1 Report

The main goal of the submitted manuscript is the 16S r RNA profiling of soil samples from pre-industrial and early industrial copper mining places in the East Harz region (Germany), with emphasis on finding interesting extremophiles and special metabolic features. Thematically, the authors continue in their recent research, changing only the site (Kohler et al., 2019) or region (Kohler et al., 2020) of collected samples. Therefore, from the point of view of originality, this is not a new topic.  Introduction is concise and clear and provides a sufficient intro to the topic as well as to the goals of the manuscript. Material and Methods are well described and allow the experiments to be repeated. The results provide an overview of the amount of laboriously obtained data, which is sometimes detrimental because one is easily lost in a flood of otherwise interesting and important facts. From this point of view, I consider the analysis and visualization of data to be the weakest part of the manuscript. I think the authors did not make full use of the potential of the enormous amount of interesting data available to them. Regarding the data analysis, it is important to divide the data into groups according to the locality where they were obtained (pre-industrial mine, slag deposit, mine dump, industrial mine dump, reference from limestone substrate, reference from acid soil), and/or according to pH (conductivity). Data analysis to be built only on the basis of the sample designation without the addition of other meaningful information (locality, pH, conductivity) loses its scientific significance. Therefore, Figures 1a - 7 should at least contain information on locality, pH or conductivity together with the sample designation. However, in the beginning, it is needed to compare the pH (conductivity) of the individual six different areas to see the data variability (e.g., box plot), whether it makes any sense to compare on the basis of pH (conductivity). Otherwise, it is not clear what is the difference between case and control and what should be considered as an extremophile or special metabolic feature.  I consider it crucial (after pH/conductivity analysis of six different locations) to compare samples composition within one of six selected groups (pre-industrial mine, slag deposit, etc.) and among all groups. From this comparison, it will be possible to formulate scientifically more relevant conclusions, which are now very general.  Moreover, from the Results and Conclusions, it is not clear what is the added value of the manuscript compared to the previous ones and why it is worthy of publishing. To state that the microbial composition of different regions differs is possible even without 16S r RNA profiling. Therefore, for example, the results in Table 2 would certainly deserve wider discussion, why these findings are so unique, and what conclusions can be drawn from these results. In conclusion, I would endorse the manuscript for publication, but only after improvement in data analysis in Results, as well as a broader discussion regarding the stated goals (finding of extremophiles and special metabolic features), which would lead to more relevant and interesting conclusions.

Kohler et al. 2019 SN Applied Science 1 839

Kohler et al. 2020 Community Ecology 21 107-120

Author Response

The main goal of the submitted manuscript is the 16S r RNA profiling of soil samples from pre-industrial and early industrial copper mining places in the East Harz region (Germany), with emphasis on finding interesting extremophiles and special metabolic features. Thematically, the authors continue in their recent research, changing only the site (Kohler et al., 2019) or region (Kohler et al., 2020) of collected samples. Therefore, from the point of view of originality, this is not a new topic.  Introduction is concise and clear and provides a sufficient intro to the topic as well as to the goals of the manuscript. Material and Methods are well described and allow the experiments to be repeated. The results provide an overview of the amount of laboriously obtained data, which is sometimes detrimental because one is easily lost in a flood of otherwise interesting and important facts. From this point of view, I consider the analysis and visualization of data to be the weakest part of the manuscript. I think the authors did not make full use of the potential of the enormous amount of interesting data available to them. Regarding the data analysis, it is important to divide the data into groups according to the locality where they were obtained (pre-industrial mine, slag deposit, mine dump, industrial mine dump, reference from limestone substrate, reference from acid soil), and/or according to pH (conductivity).

Answer:

The samples have been divided into 7 groups. For overview, the new table 2 was added. The group designation was additionaly included in table 1. The data analysis and discussion was extended to distinguish between these groups in different paragraphs of the results- and-discussion part and in figure captions.

Table 2: Groups of soil samples

  1. A) pre-industrial mines, pH < 7.5
  2. B) pre-industrial mines, pH > 7.5
  3. C) mine dump, early 19th century, pH>8
  4. D) slag deposit, pH 5.57
  5. E) industrial mine dump, 7.66
  6. F) reference samples from lime stone substrate, pH>6.5
  7. G) reference samples from acid soil substrates, pH<5

However, in the beginning, it is needed to compare the pH (conductivity) of the individual six different areas to see the data variability (e.g., box plot), whether it makes any sense to compare on the basis of pH (conductivity). Otherwise, it is not clear what is the difference between case and control and what should be considered as an extremophile or special metabolic feature.  I consider it crucial (after pH/conductivity analysis of six different locations) to compare samples composition within one of six selected groups (pre-industrial mine, slag deposit, etc.) and among all groups. From this comparison, it will be possible to formulate scientifically more relevant conclusions, which are now very general. Data analysis to be built only on the basis of the sample designation without the addition of other meaningful information (locality, pH, conductivity) loses its scientific significance. Therefore, Figures 1a - 7 should at least contain information on locality, pH or conductivity together with the sample designation.

Answer:

Group designations (A-G)  are added to figures or figure captions. Other data have been added in figure captions:

Fig. 1

  1. a) Percentage of the most abundant phyla in the 19 investigated samples from formerly copper mining areas (No. 1 – 19, groups A-E) and the reference samples (No. 20 – 25, groups F and G ). b) Domination of single OTUs: Percentages of the total number of reads for four single OTUs with high abundance in the samples 5 (Wiederstedt, group B, pH 7.72), 7 (Welfesholz, group C, pH 8.24), 8 (Welfesholz, group C, pH 8.78), 10 (Burgörner, group C, pH 8.85), 11 (Burgörner, group C, pH 8.28) and 12 (Hettstedt, group C, pH 8.01)

Fig. 2

Highly abundant taxonomic groups (total number of reads per sample): a) Acidobacteria, b) Aci-dimicrobiia; The numbers on the x-axis are related to the sample numbers of table 1

Fig. 3

Distribution of abundant orders of Acidobacteria in 5 selected samples from ancient mining areas (No. 3 Wimmelburg, slag deposit, pH 5.57), 8 (Welfesholz, group C, pH 8.78), 10 (Burgörner, group C, pH 8.85), 11 (Burgörner, group C, pH 8.28), 16 (Uftrungen, Group A, pH 6.23), from industrial mine dump (No. 19, Nienstedt, pH 7.66, extraordinary high electrical conductivity) and four reference samples (limestone substrate: No. 20 and 22 (group F), acid soil: No. 24, 25 (group G))

Fig. 5

Distribution of abundant OTUs of Acidimicrobiia (phylum Actinobacteria) in 5 selected samples from ancient mining areas (No. 3 Wimmelburg, slag deposit, pH 5.57), 8 (Welfesholz, group C, pH 8.78), 10 (Burgörner, group C, pH 8.85), 11 (Burgörner, group C, pH 8.28), 16 (Uftrungen, Group A, pH 6.23), from industrial mine dump (No. 19, Nienstedt, pH 7.66, extraordinary high electrical con-ductivity) and four reference samples (limestone substrate: No. 20 and 22 (group F), acid soil: No. 24, 25 (group G))

Fig. 6

Distribution of selected abundant orders of Actinobacteria (without Acidimicrobiia) in 5 selected samples from ancient mining areas (No. 3 Wimmelburg, slag deposit, pH 5.57), 8 (Welfesholz, group C, pH 8.78), 10 (Burgörner, group C, pH 8.85), 11 (Burgörner, group C, pH 8.28), 16 (Uftrungen, Group A, pH 6.23), from industrial mine dump (No. 19, Nienstedt, pH 7.66, extraor-dinary high electrical conductivity) and four reference samples (limestone substrate: No. 20 and 22 (group F), acid soil: No. 24, 25 (group G))

Fig. 7

Distribution of abundant OTUs of Chloroflexi in 5 selected samples from ancient mining areas (No. 3 Wimmelburg, slag deposit, pH 5.57), 8 (Welfesholz, group C, pH 8.78), 10 (Burgörner, group C, pH 8.85), 11 (Burgörner, group C, pH 8.28), 16 (Uftrungen, Group A, pH 6.23), from industrial mine dump (No. 19, Nienstedt, pH 7.66, extraordinary high electrical conductivity) and four reference samples (limestone substrate: No. 20 and 22 (group F), acid soil: No. 24, 25 (group G))

Fig. 8

Abundances of nine selected OTUs on ancient mining sites, absent in the reference samples from acidic soil (No. 23 – 25, group G)

Fig. 9

Abundances of nine selected OTUs with acidophilic character and comparatively high representation in the reference samples from acid soil (No. 23 – 25, group G)

Moreover, from the Results and Conclusions, it is not clear what is the added value of the manuscript compared to the previous ones and why it is worthy of publishing. To state that the microbial composition of different regions differs is possible even without 16S r RNA profiling. Therefore, for example, the results in Table 2 would certainly deserve wider discussion, why these findings are so unique, and what conclusions can be drawn from these results. In conclusion, I would endorse the manuscript for publication, but only after improvement in data analysis in Results, as well as a broader discussion regarding the stated goals (finding of extremophiles and special metabolic features), which would lead to more relevant and interesting conclusions.

Answer:

The data analysis and discussion was improved and extended based on the subdivision in the groups of the new table 2:

In the results-and-discussion section:

Particularly high contents in the group of four frequent OTUs (Sphingomonas, IMCC26256, Ralstonia and Ilumatobacteraceaea) are observed in the samples No. 5, 7, 8, 10, 11 and 12 (belonging mainly to sample group C), which differ from all other samples by these types (Fig. 1b). This difference between samples seems to be related to some extent to the pH of the soil, because the samples No. 7 – 12 (all group C) showed the highest pH values (see table 1). It should however be noted that the samples No. 9 is not included in this group, despite its pH is also above 8.  The abundances of Sphingomonas in some samples could be of special interest because their metabolic potential is important for the whole community, for example by production of phytohormones as gibberellines and indol acetic acid [22], which might also be insteresting for future improvement of soil fertility and remediation by means of biotechnological products.  …

For example, sample 16 supplied in 26% of the reads the OTU Candidatus Ud. (Chthoniobacteraceae) and sample 3 supplied in 36% of reads two OTUs of Ktedonobacteraceae. From this table, it can be seen clearly, that the samples No 5, 7, 8, 10, 1, 12 (mostly group C) are distinguished from the other samples, which are closer related to the reference samples from lime stone substrates (Group F). The table shows also clearly the special situation of samples 3 and 19.

In sample 19, many OTUs were found with mediate or high read numbers, which are very rare or completely absent in all other samples (see supplementary table S2). The special soil communities of this sampling site could be explained by the special situation of this place. On the one hand, this large mine dump was operated in the 20th century, what means it is closed since some decades, only. This is in strong contrast to the small mines and dumps from the early industrial and the pre-industrial era, which rest since several centuries. The place (No 19) has nearly no vegetation and, therefore it was surprising to see the large spectrum of different OTUs in the NGS data. In addition, this place showed the highest electrical conductivity of soil speaking for a high content of soluble salts. This fact explains the high content of halophilic types, but let expect, in addition, another character of microbial community as whole.

…critical environmentally hazardous substances like aromatic and polycyclic hydrocarbons (table 2). The special types have different occurrence in the sample groups. Thus, the polyvinylalcohol-degrading Povalibacter, the halophilic bacteria Lewinella and Halobacteriovorax were found in group A (pre-industrial mines), whereas poymer-degrading acidophilic organisms among them Thaumarchaeota appeared in the soil of the slag deposit area (D).  

A noticeable high number of reads (626) was also found for Psychroflexus, which is famous for its special ability to produce eicosanopentaeonic acid, a polyunsaturated fatty acid [48]. Despite the occurrence of a lot of halophilic organisms in this sample, the content of psychrophilic genera confirms the special character of the related soil bacterial commu-nity. The coincidence of salt content, appearance of psychrophiles and an high probability of temporary exposure to heavy metals from the dump erosion support the formation of soil microbioms of special compositions containing types, from which special metabolic features could be expected, for example related to metalloenzymes.

In the conclusions following addition has made:

…Some of the strains belonging to the genera found produce different types of pigments or other special metabolites or a polyunsaturated fatty acid, for example. The differences in the compositions of soil microbial communities and the abundances of genera known for extremophilic behavior are not related to soil pH exclusively and salt content. In addition, components of soil bacteria were found in the samples indicating that different sampling sites can be valuable for the search of new strains with interesting metabolic properties.

Reviewer 2 Report

This paper described on culture independent approach to analyse microbiota in soil communities of former copper mining sites.

Major:

Abstract:

This content seems mediocre, so please give a concrete example of the remarkable result.

Materials and Methods:

Please show a map etc. to give the reader a more concrete image of the difference in origin of each sample. If possible, please provide information such as relationships with historical events and temperature changes throughout the year in each sampling site. Also, please give strategic explanations of your research when adopting these samples.

Results and discussion:

Please provide a little more in-depth consideration of the following:

1) Reason for No. 19 peculiarity in microbial flora.

2) Reason for the peculiarity of No. 19 as a source of useful gene resources.

Minor:

Most figures do not have vertical and horizontal labels.

The reviewer thinks that the number of references published in the last three years is small.

Author Response

Abstract: This content seems mediocre, so please give a concrete example of the remarkable result.

Answer:

The abstract was improved and some details on proved special OTUs have been added:

…Among these OTUs, several DNA sequences are related to bacteria which are reported to show the ability to metabolize special substrates. Special OTUs absent in reference samples from lime stone substrates, among them Thaumarchaeota were present in the soil group from ancient mines with pH>7. In contrast, acidophilic types have been found in a sample from a copper slag deposit, for example the polymer degrading bacterium Granulicella and Acidicaldus, which is thermophilic, too. Soil samples of the group of pre-industrial mines supplied some less abundant interesting OTUs as the polymer-degrading Povalibacter and the halophilic Lewinella and Halobacteriovorax. A particular high number of special bacteria were found at an industrial copper mine dump among them many halophilic and psychrophilic types. In summary, the results show …

Materials and Methods: Please show a map etc. to give the reader a more concrete image of the difference in origin of each sample. If possible, please provide information such as relationships with historical events and temperature changes throughout the year in each sampling site. Also, please give strategic explanations of your research when adopting these samples.

Answer:

For giving an image of the sampling sites, a map was added to the supplementary material. Further, a paragraph in Materials-and-Methods was added for explaining the idea behind the selection of sampling sites:

The idea behind the selection of soil samples was to search for special bacterial types related to the past of sampling sites connected with different ages of the mining activities. Thus, small mining places, in particular from the early mining activities in the 15th - 18th century (Uftrungen, Rodishain, Pölsfeld, Hergisdorf, Wolferode and Wiederstedt), early industrial mining sites (1st half of 19th century; Burgörner, Hettstedt, Welfesholz) and two industrial places (20th century) have been choosen.

… The reference samples have been choosen from a sample collection of different parts of Thuringia in order to have different regions and different geological situations. The samples from line stone substrates (pH > 7.5) came from northwest Thuringia (Haynrode), North Thuringia (Burgwenden) and from the Saale valley (Kahla). The samples from acid substrate soils were taken from southwest Thuringia (Northern Rhön, Völkershausen, basalt), South Thurigina (Suhl, sand stone) and Northeast Thurigina (Rastenberg, sand stone).

Results and discussion: Please provide a little more in-depth consideration of the following:

1) Reason for No. 19 peculiarity in microbial flora.

Answer:

Following section was introduced for clearification:

… In sample 19, many OTUs were found with mediate or high read numbers, which are very rare or completely absent in all other samples (see supplementary table S2). The special soil communities of this sampling site could be explained by the special situation of this place. On the one hand, this large mine dump was operated in the 20th century, what means it is closed since some decades, only. This is in strong contrast to the small mines and dumps from the early industrial and the pre-industrial era, which rest since centuries . The place (No 19) has nearly no vegetation and, therefore it was surprising to see the large spectrum of different OTUs in the NGS data. In addition, this place showed the highest electrical conductivity of soil speaking for a high content of soluble salts. This fact explains the high content of halophilic types, but let expect, in addition, another character of microbial community as whole.

2) Reason for the peculiarity of No. 19 as a source of useful gene resources.

Answer:

Some arguments are given in the text block above. In addition, following text block was added:

… A noticeable high number of reads (626) was also found for Psychroflexus, which is famous for its special ability to produce eicosanopentaeonic acid, a polyunsaturated fatty acid [48]. Despite the occurrence of a lot of halophilic organisms in this sample, the content of psychrophilic genera confirms the special character of the related soil bacterial community. The coincidence of salt content, appearance of psychrophiles and a high probability of temporary exposure to heavy metals from the dump erosion support the formation of soil microbioms of special compositions containing types, from which special metabolic features could be expected, for example related to metalloenzymes.

Minor:

Most figures do not have vertical and horizontal labels.

The missed labels are added now (Figs. 1,2,4,8).

The reviewer thinks that the number of references published in the last three years is small.

There have been added, some recent important references:

  1. Hedrich, S.; Schippers, A. Distribution of acidophilic microorganisms in natural and man-made acidic environments. Curr. Issues Mol. Biol. 2021, 40, 25-47.
  2. Ratzke, C.; Barrere, J.; Gore, J. Strength of species interactions determines biodiversity and stability in microbial communities. Nature Ecol. Evol. 2020, 4, 376.
  3. Purohit, J.; Chattopadhyay, A.; Teli, B. Metagenomic exploration of plastic degrading microbes for biotechnological applications. Curr. Genomics 2020, 21, 253-270.
  4. Asaf, S.; Numan, M.; Khan, A.L.; Al-Harrasi, A. Sphingomonas: from diversity and genomics to functional role in environmental remediation and plant growth. Crit. Rev. Biotechnol. 2020, 40, 138-152.

Reviewer 3 Report

The topic of the paper and its assumptions and research object are very interesting. Unfortunately, due to gross negligence in the presentation of results I do not recommend it for publication in Microorganisms.

Major comments:

  1. Presentation of results is at a very low level. The simplest excel charts were made, without axis captions, without detailed explanations, a picture of an excel table was inserted... And it is possible to make heatmaps, heatmaps with clustering, extended pie diagram (e.g. Venn diagrams, phylogenetic trees, co-occurrence plot etc.), or to tabulate data in case of problems.
  2. importantly, there are no statistics. Neither between samples, whether these results are statistically significant at all, nor correlations with soil parameters. No correlations, PCA, PCoA. Nothing.

3) In fact, there is not even any information on whether samples were taken in replicates or whether analyses were performed in replicates.

  1. Too little information about the soils used in the study. pH and EC are definitely not enough.
  2. The discussion is poor.
  3. The purpose of the study states " …investigated for their possible relevance in searching for interesting extremophiles and special metabolic features…", but in the results we have only one table concerning metabolic features. This is not enough to conclude that the aim of the study was fulfilled. Moreover, the table is based on literature data, yet there are bioinformatic tools available to assess potential metabolic functions in the analysed samples.

Other comments:

L25: co to znaczy „references samples”?

L35: genetic profiling? Not really. Too exaggerated a slogan in relation to 16S rRNA analysis. Sequencing or NGS will be enough..

L81-84: How, when were the samples taken? In what quantity? From what depth? Add climatic conditions (general rainfall and temp.), soil type. Was it taken in replicates?

L89-92: „references samples” is too lofty a wording. Rather simply controlling.

Table 1: Please format it according to the template. Something has gone awry in the title lines - please correct. A unit should be added to electrical conductivity. If sample numbers (No) are given throughout the paper as their designation (on graphs etc.) then caption them in the table not as No, but as the sample symbol. And then refer to the table as information about the samples. In the table, the numerical notation for EC should be standardised - either 1 decimal place or none everywhere.

L101-104: add referencees for primers

L105, 115, 118: correct the entry of the unit, i.e. pmoL µL-1 etc.

L93-124: What NGS analysis was done, on what equipment? Explain the term NGS. Clearly describe the sequencing step.

Was the analysis done in replicates? Isolation, sequencing? Was any statistical analysis done?

L128-134: what parameters were adopted for processing the results?              

L125, 129: please correct the references - here they should be given numerically and the reference and description placed normally in the literature list.

Figure 1: no unit and title on Y-axis, no title on X-axis; no explanation of what 'other' means in the graph. Graph A is not very readable (too mottled).  In Fig. 1B it is worth making the scale more dense, because the values of the lowest bars cannot be estimated. Statistics?

Figure 2: No unit and title on Y axis, no title on X axis; Statistics?

Figure 3, 5, 6, 7: What is the unit on the charts? What are the 'others'?

L193-194: There is nothing in figure 4 that confirms the corelation of these bacteria with pH or EC. Where did this information come from? There is no statistical analysis in the paper at all. We do not know if the differences in any of the graphs are statistically significant.

Figure 4: No unit and title on Y axis, no title on X axis; Statistics?

Figure 8, 9: No unit and title on Y axis, no title on X axis; Statistics? Caption does not explain what is on graphs a-c.

The discussion is very poor. I recommend expanding it.

L300-307: references?

L307: correct „(Table 2)”.

Table 2: this is a screenshot of an excel table. This is at least inappropriate for a scientific paper. Please format the table correctly or present the results in a different way.

Author Response

The topic of the paper and its assumptions and research object are very interesting. Unfortunately, due to gross negligence in the presentation of results I do not recommend it for publication in Microorganisms.

Major comments:

Presentation of results is at a very low level. The simplest excel charts were made, without axis captions, without detailed explanations, a picture of an excel table was inserted... And it is possible to make heatmaps, heatmaps with clustering, extended pie diagram (e.g. Venn diagrams, phylogenetic trees, co-occurrence plot etc.), or to tabulate data in case of problems.

Answer:

The general level of paper was improved by a lot of modifications and new discussions.  Figure captions have been improved and axis labels have been added.

importantly, there are no statistics. Neither between samples, whether these results are statistically significant at all, nor correlations with soil parameters. No correlations, PCA, PCoA. Nothing.

Answer:

Groups of samples are compared in the revised manuscript. Correlation diagrams and PCAs are added in three new figures.

3) In fact, there is not even any information on whether samples were taken in replicates or whether analyses were performed in replicates.

Too little information about the soils used in the study. pH and EC are definitely not enough.

Answer:

A map was added to the supplementary material in order to give information about the distribution of sampling sites.

The discussion is poor. The purpose of the study states " …investigated for their possible relevance in searching for interesting extremophiles and special metabolic features…", but in the results we have only one table concerning metabolic features. This is not enough to conclude that the aim of the study was fulfilled. Moreover, the table is based on literature data, yet there are bioinformatic tools available to assess potential metabolic functions in the analysed samples.

Answer:

Indeed, we have not made own bioinformatic studies on metabolic features, but important aspects of possibly expected metabolic features of bacteria are discussed in the frame of literature. The related discussions have been improved and extended in the revised manuscript.

It should however be noted that the samples No. 9 is not included in this group, despite its pH is also above 8.  The abundances of Sphingomonas in some samples could be of special interest because their metabolic potential is important for the whole community, for example by production of phytohormones as gibberellines and indol acetic acid [22], which might also be insteresting for future improvement of soil fertility and remediation by means of biotechnological products. The bacterial communities …

In sample 19, many OTUs were found with mediate or high read numbers, which are very rare or completely absent in all other samples (see supplementary table S2). The special soil communities of this sampling site could be explained by the special situation of this place. On the one hand, this large mine dump was operated in the 20th century, what means it is closed since some decades, only. This is in strong contrast to the small mines and dumps from the early industrial and the pre-industrial era, which rest since several centuries. The place (No 19) has nearly no vegetation and, therefore it was surprising to see the large spectrum of different OTUs in the NGS data. In addition, this place showed the highest electrical conductivity of soil speaking for a high content of soluble salts. This fact explains the high content of halophilic types, but let expect, in addition, another character of microbial community as whole.

A noticeable high number of reads (626) was also found for Psychroflexus, which is famous for its special ability to produce eicosanopentaeonic acid, a polyunsaturated fatty acid [52]. Despite the occurrence of a lot of halophilic organisms in this sample, the content of psychrophilic genera confirms the special character of the related soil bacterial commu-nity. The coincidence of salt content, appearance of psychrophiles and an high probability of temporary exposure to heavy metals from the dump erosion support the formation of soil microbioms of special compositions containing types, from which special metabolic features could be expected, for example related to metalloenzymes.

Several of these types have been reported to be able to degrade special substances, for example agar, gellan-gum, salicylate, molinate, alkanes, synthetic polymers like pol-yvinylalcohol and critical environmentally hazardous substances like aromatic and pol-ycyclic hydrocarbons (table 2). The special types have different occurrence in the sample groups. Thus, the polyvinylalcohol-degrading Povalibacter, the halophilic bacteria Lew-inella and Halobacteriovorax were found in group A (pre-industrial mines), whereas poymer-degrading acidophilic organisms among them Thaumarchaeota appeared in the soil of the slag deposit area (D).                                

Other comments:

L25: co to znaczy „references samples”?

A set of six samples from other regions and without copper mining activities was chosen in order to distinguish pH-related effects and place- or mining-related differences in the soil bacterial community. Following text blocks were added in the Materials- and-Methods section for explaining the motivation behind the selection of samples:

The idea behind the selection of soil samples was to search for special bacterial types related to the past of sampling sites connected with different ages of the mining activities. Thus, small mining places, in particular from the early mining activities in the 15th - 18th century (Uftrungen, Rodishain, Pölsfeld, Hergisdorf, Wolferode and Wiederstedt), early industrial mining sites (1st half of 19th century; Burgörner, Hettstedt, Welfesholz) and two industrial places (20th century) have been choosen.

…six other samples are included as references which origin from forest-covered areas of ancient ramparts: three from lime stone substrates (No 20-22) and three from acid mineral substrates (quartzite and sandstone, No 23-25). The reference samples have been choosen from a sample collection of different parts of Thuringia in order to have different regions and different geological situations. The samples from line stone substrates (pH > 7.5) came from northwest Thuringia (Haynrode), North Thuringia (Burgwenden) and from the Saale valley (Kahla). The samples from acid substrate soils were taken from southwest Thuringia (Northern Rhön, Völkershausen, basalt), South Thurigina (Suhl, sand stone) and Northeast Thurigina (Rastenberg, sand stone).

L35: genetic profiling? Not really. Too exaggerated a slogan in relation to 16S rRNA analysis. Sequencing or NGS will be enough..

Answer:

The keyword was substituted (NGS).

L81-84: How, when were the samples taken? In what quantity? From what depth? Add climatic conditions (general rainfall and temp.), soil type. Was it taken in replicates?

Answer:

Following text block was modified and extend for explanation:

The sample material (No 1-19) is not any kind of typical natural soil, but surface material from the mine dumps, what means material which was brought from the ground to the surface during the site-related mining phases. The soil material was taken during dry weather periods. For sampling vegetation-free spots of the surface have been used.

The samples are originating from a sampling series without the collection of replicates.

L89-92: „references samples” is too lofty a wording. Rather simply controlling.

I would prefer to keep the terminus “Reference samples” in the manuscript because they are more used for a comparison of different sites and soils than for a typical “control”. Following modification was made in the text:

 Beside this sample, six other samples are included as references (control samples) which origin from forest-covered areas

Table 1: Please format it according to the template. Something has gone awry in the title lines - please correct. A unit should be added to electrical conductivity. If sample numbers (No) are given throughout the paper as their designation (on graphs etc.) then caption them in the table not as No, but as the sample symbol. And then refer to the table as information about the samples. In the table, the numerical notation for EC should be standardised - either 1 decimal place or none everywhere.

L101-104: add referencees for primers

The supplier is referred in the materials-and-methods part.

L105, 115, 118: correct the entry of the unit, i.e. pmoL µL-1 etc.

The units are corrected, now.

L93-124: What NGS analysis was done, on what equipment? Explain the term NGS. Clearly describe the sequencing step.

The sequencing was done by GATC Biotech (Konstanz, Germany) and Eurofins (Ebersberg, Germany) by next generation sequencing on an Illumina HiSeq.

Was the analysis done in replicates? Isolation, sequencing? Was any statistical analysis done?

Quality of DNA purification was checked by gel electrophoreses after each PCR step.Replcates had not be done. For improvement PCA and correlationsof sequence analyses  have been added to the manuscript, now.

L128-134: what parameters were adopted for processing the results?             

Following information was added:

…data files from the NGS analyses have firstly been converted from fastq file format to fasta file format by use of the fastq-to-fasta converter tool “phred33 conversion” (MR DNA Lab). The reads are aligned by SILVA Incremental Aligner (SINA SINA v1.2.10 for ARB SVN (revision 21008)) against the SILVA SSU rRNA SEED and quality controlled (applying minimal identity criteria of 98 %). Reads with a low alignment quality, less than 50 aligned nucleotides and reads with more than two percent of ambiguities, or two percent of homopolymers, respectively, are not processed. For classification a BLAST search with the database SILVA Ref (release 132) was applied. For all analyses …

L125, 129: please correct the references - here they should be given numerically and the reference and description placed normally in the literature list.

The references are given numerically, now.

Figure 1: no unit and title on Y-axis, no title on X-axis; no explanation of what 'other' means in the graph. Graph A is not very readable (too mottled).  In Fig. 1B it is worth making the scale more dense, because the values of the lowest bars cannot be estimated. Statistics?

Fig. 1 is improved now.

Figure 2: No unit and title on Y axis, no title on X axis; Statistics?

Fig. 2 is improved now.

Figure 3, 5, 6, 7: What is the unit on the charts? What are the 'others'?

The data are explained in the figure captions, now.

L193-194: There is nothing in figure 4 that confirms the corelation of these bacteria with pH or EC. Where did this information come from? There is no statistical analysis in the paper at all. We do not know if the differences in any of the graphs are statistically significant.

New figures 2-4  are added in order to illustrate correlations in the now defined sample groups. The discussion was extended.

Figure 4: No unit and title on Y axis, no title on X axis; Statistics?

Fig. 4 is improved, now.

Figure 8, 9: No unit and title on Y axis, no title on X axis; Statistics? Caption does not explain what is on graphs a-c.

The data are explained in the figure captions, now.

The discussion is very poor. I recommend expanding it.

The discussion is extend and improved, now.

L300-307: references?

Additional references are added.

L307: correct „(Table 2)”.

The tables are improved.

Table 2: this is a screenshot of an excel table. This is at least inappropriate for a scientific paper. Please format the table correctly or present the results in a different way.

The table (new number: table 3) is added as doxc-file attachement, now.

Round 2

Reviewer 1 Report

I acknowledge that certain changes have been made (a division of samples into groups according to pH, PCA), but they are still not sufficient for the manuscript to be eligible for publication. Adding more graphs to the manuscript made it even more confusing than before. Authors should select a maximum of 6 plots with detailed descriptions, relevant statistics, and strong evidence of the main idea they wanted to present. The rest of the results should be added to the Supplementary material. In a flood of data and a large number of simple graphs, which have only a minimal informative value in terms of the main idea, the reader is easily lost and misses the point of the manuscript. Unfortunately, regarding the new information related to the control samples (lines 101-107), it turned out that the reference samples were not selected correctly. On the new lines 90-91, the authors state that "The idea behind the selection of soil samples was to search for special bacterial types related to the past of sampling sites connected with different ages of the mining activities." From this point of view, the correct reference must be from the same geological substrate as the examined sample but without any mining activity. This applies to all samples. In the context of this idea, even this amount of data does not give any scientifically relevant output. It is just a simple description of a lot of results from a lot of places. Therefore, either the authors complete the correct references or reconsider and clearly describe the main scientific goal and output of the manuscript (not just a simple listing of results). In this form, the manuscript is certainly still not suitable for publication. 

Lines 86-87: In the Supplementary material, the map is still missing.

Author Response

Answer to reviewer comments,

At first, I like to thank cordially for all of your efforts and recommendations for improvement of the manuscript.

But in the recent situation, I have to explain our original intention concerning this manuscript: It was invited as a contribution to a special on extremophiles. Therefore, we re-analyzed archived NGS data from samples from ancient mining sites with the focus on relations to extremophiles. No more was intended. After the first review round, we tried to improve the manuscript. This was a compromise between a trial to respond to your very valuable comments, our recent state of data and our original intention. After your second comments, we had decided to withdraw the manuscript, because we are not able to improve it in the recommended way.

Now, the editor encouraged us to submit a new revised version.  In this situation, I have to apologize for responding with a modified manuscript, but not with a complete rewritten contribution following your advices. We have reduced the number of figures and shifted other figures and data into the supplementary.

I would be really grateful for your recommendation for decision to publish the submitted material or to reject it or to withdraw the manuscript, finally.

Reviewer 2 Report

The paper described on microbial communities of former copper mining sites was revised.

  1. Although there is a description of a map exhibiting the sampling site (P2 in the paper), I can't see a map showing the geographical location of the sampling site in the supplementary material.
  2. Contents of Table 2 should be explanations of Table 1.
  3. Since there are too many data in the main text and it seems that the current presenting manner dilutes very important thing. Therefore, this reviewer recommends organizing the all the information and leave only the important information in the main body, and then make it Supplementary Figs.
  4. The overall structure of Figs is difficult to consider what can be said from that. Please think about ways to make it easier for readers to consider, such as making it possible to understand the classification of Table 2 as shown in Fig. 7.
  5. There is no description on the determined sequences in this study at “Data availability statement”.

Author Response

(The authors gave the same response as above.)

Reviewer 3 Report

  • Table 1: Please format it according to the template - without horizontal lines.
  • L116-120: add referencees for primers – not the supplier, but on what basis these and not other primers were chosen. Literature source.  
  • Figure 1: lack of explanation of what 'other' means in the graph. Why on fig. A have 25 samples and in fig. B 24?
  • Figure 2: % for PC1 and PC2? Please add an intersection at 0.0 to make the graphs more readable. There should be full stops instead of commas in the unit notation.
  • Figure 3: % for PC1 and PC2? Please add an intersection at 0.0 to make the graphs more readable. There should be full stops instead of commas in the unit notation. Figure A has unexplained symbols. A list of OTUs is given, but are they in the diagram? Which ones are which? Figure B has no caption for the legend (groups).
  • Figure 4: Please add an intersection at 0.0 to make the graphs more readable. There should be full stops instead of commas in the unit notation. There is no caption for the legend (groups).
  • Figure 6, 8, 9, 10: The unit should be placed on the charts (%). And 'others' explained under the figure. The font is too small. Graphs are numerous, but they do not enrich the paper in my opinion.
  • Figure 12: No unit and title on Y axis, no title on X axis.
  • Table 2: Please format it according to the template - without horizontal lines. The pH was not determined in repetition either? To make a statistic? Besides, the division of samples into groups to which you keep referring in the results should be mentioned in the materials and discussed there.
  • The discussion is still very poor. I recommend expanding it. The idea is not to add more text, but to discuss more results with other papers. The authors write that they have improved the discussion, but in reality we only have a few fragments of it: L186-190, L278-282, L317-319, L350-355, L380-396. The rest is the presentation of the results. If authors find it difficult to maintain the balance between the discussion of results and the discussion, I suggest separating these parts into "results" and "discussion". Maybe then the paper will look better.
  • „The purpose of the study states " …investigated for their possible relevance in searching for interesting extremophiles and special metabolic features…", but in the results we have only one table concerning metabolic features. This is not enough to conclude that the aim of the study was fulfilled. Moreover, the table is based on literature data, yet there are bioinformatic tools available to assess potential metabolic functions in the analysed samples. Answer: Indeed, we have not made own bioinformatic studies on metabolic features, but important aspects of possibly expected metabolic features of bacteria are discussed in the frame of literature. The related discussions have been improved and extended in the revised manuscript.”  - I suggest that the wording of the objective be changed. To make it clear that this is just a discussion, literature data etc. at the moment the reader is being misled. Other than that, I did not find much information or reference to this issue in the discussion.

Author Response

(The authors gave the same response as above.)

Round 3

Reviewer 1 Report

Dear Author,

I understand your situation, but as a reviewer, I have to keep a scientific level of the manuscript. If you still insist that samples no. 20-25 are references to the mining areas, I have to reject the publication because it is not true, and a basic requirement for a scientific manuscript (correct reference) is not observed. The samples no. 20-25 are only supplement information from non-mining areas to the data from mining areas but can not be a reference to make comparable scientific conclusions because do not come from the same but non-mining substrate/area. Therefore the sentences from the Conclusions are not scientifically correct (Lines 418-423: “Even the mining sites operated in the pre-industrial period (about 16th to 18th century) and the early industrial period (first half of 19th century) supplied bacterial types that differ in their occurrence or abundance from six selected reference sites. The occurrence of some taxonomic groups of  acidophilic bacteria correlates with reference samples from forest areas on acidophilic substrates, others differ significantly from them.”). From this point of view, the main idea of the manuscript (lines 93-94: "The idea behind the selection of soil samples was to search for special bacterial types related to the past of sampling sites connected with different ages of the mining activities.") can not be fulfilled. I am sorry because I believe that the manuscript costs a lot of work, and in addition, the obtained data have great potential, but they must be used and interpreted scientifically correct. However, in this format, I have to reject the manuscript.    

Author Response

I understand your situation, but as a reviewer, I have to keep a scientific level of the manuscript. If you still insist that samples no. 20-25 are references to the mining areas, I have to reject the publication because it is not true, and a basic requirement for a scientific manuscript (correct reference) is not observed. The samples no. 20-25 are only supplement information from non-mining areas to the data from mining areas but can not be a reference to make comparable scientific conclusions because do not come from the same but non-mining substrate/area. Therefore the sentences from the Conclusions are not scientifically correct (Lines 418-423: “Even the mining sites operated in the pre-industrial period (about 16th to 18th century) and the early industrial period (first half of 19th century) supplied bacterial types that differ in their occurrence or abundance from six selected reference sites. The occurrence of some taxonomic groups of  acidophilic bacteria correlates with reference samples from forest areas on acidophilic substrates, others differ significantly from them.”).

The related sentence in the conclusions is changed, now:

Even the mining sites operated in the pre-industrial period (about 16th to 18th century) and the early industrial period (first half of 19th century) supplied bacterial types that differ in their occurrence or abundance from six selected comparative samples from non-mining sites. The occurrence of some taxonomic groups of acidophilic bacteria correlates with comparative samples from forest areas …

For avoiding confusion concerning the term “reference samples”, it was substituted by “comparative samples” in other parts of the manuscript.

From this point of view, the main idea of the manuscript (lines 93-94: "The idea behind the selection of soil samples was to search for special bacterial types related to the past of sampling sites connected with different ages of the mining activities.") can not be fulfilled. I am sorry because I believe that the manuscript costs a lot of work, and in addition, the obtained data have great potential, but they must be used and interpreted scientifically correct. However, in this format, I have to reject the manuscript.

The related sentence was modified in order to avoid a misunderstanding concerning the motivation of the work:

The idea behind the selection of soil samples was to search for special bacterial types related to mining activities in the past and compare them with some samples from non-mining places.

Reviewer 2 Report

The paper described in the microbiol communities of former copper mining sites in Germany has been revised.

  • This reviewer understood that this research is very interesting, but the list of data is too large to understand the essence. This reviewer recommends that the authors organize the information once again, devise a way of showing it to explain the core part of this research accurately, and post only the data you really need as Figs in the text.

  • Although the description in the “Abstract” seemed to be improved, the authors should explain how “special” OTUs or bacteria.

  • There is no description how the authors created PCA (please described the name of alogism used) and Figure 4.

  • Although Table 1 was cited in the text, it seemed that Table 1 was eliminated. Table 1 should be present. Definition for group (A-G) should be described in Table 1. If Table 2 was eliminated.

Author Response

The paper described in the microbiol communities of former copper mining sites in Germany has been revised.

This reviewer understood that this research is very interesting, but the list of data is too large to understand the essence. This reviewer recommends that the authors organize the information once again, devise a way of showing it to explain the core part of this research accurately, and post only the data you really need as Figs in the text.

The manuscript was already re-organized for reconsidering recommendations of reviewers. The “results and discussion” part (3) is now subdivided into two main sections only:

3.1. Phyla and dominant taxonomical units

3.2. Abundances of taxonomical groups related to extremophilic characters

The following sections are marked by (3rd order) subheadings 3.2.1. - 3.2.5.

Although the description in the “Abstract” seemed to be improved, the authors should explain how “special” OTUs or bacteria.

“Special” was partially substituted

In the abstract:

  Some OTUs absent in comparative samples from lime stone substrates,…

A particular high number of bacteria (OTUs) which had not been detected in other samples were found …

There is no description how the authors created PCA (please described the name of alogism used) and Figure 4.

 For PCA, Matlab was applied. Following sentence is added on the end of section 2:

PCA was performed by using the Matlab standard procedure.

Although Table 1 was cited in the text, it seemed that Table 1 was eliminated. Table 1 should be present. Definition for group (A-G) should be described in Table 1. If Table 2 was eliminated.

Table 1 is added now.

Reviewer 3 Report

If this is the case, the paper should have indicated from the outset that it relates to studies that have been carried out in the past, and only in a different aspect are they recognised. This should be emphasised both in the introduction and later in the discussion of results. And maybe also somehow in the title.
Since these samples were analysed earlier, it is possible to write for what purpose and to refer to publications about them or to give some information in the text. Because at the moment the aim of the paper indicates that you did research on extremophiles and that is it. That is why I have so many comments on it.
The whole methodology suggests that you did the research from the beginning - from sample selection, through isolation and then NGS analysis. If you have already only the finished NGS results (this is how I understood it now), then you should describe it clearly and explicitly. And focus on the data analysis itself.

In line 89 you refer to a table 1, which you have removed from the text.

Author Response

If this is the case, the paper should have indicated from the outset that it relates to studies that have been carried out in the past, and only in a different aspect are they recognised. This should be emphasised both in the introduction and later in the discussion of results. And maybe also somehow in the title.

The introduction is modified, now:

Therefore, here 16S rRNA profiling data of soil samples from East Harz copper mining places are analyzed and partially re-analyzed, resp., for their possible relevance in searching for interesting extremophiles.

The title was modified:

Extremophiles in Soil Communities of Former Copper Mining Sites of the East Harz Region (Germany) Reflected by re-analyzed 16S rRNA Data

Since these samples were analysed earlier, it is possible to write for what purpose and to refer to publications about them or to give some information in the text. Because at the moment the aim of the paper indicates that you did research on extremophiles and that is it. That is why I have so many comments on it.

Originally, we had been mainly interested in looking for heavy metal-tolerant soil bacterial strains and site-specific patterns. For explaining the original motivation, following additional sentence is added to the introduction and related to Ref. [15]:

The soil samples had been collected, originally, for searching for heavy metal-tolerant bacterial strains, but then found to be of interest with respect to halophilic, acidophilic and thermophilic bacteria, too [15].

The whole methodology suggests that you did the research from the beginning – from sample selection, through isolation and then NGS analysis. If you have already only the finished NGS results (this is how I understood it now), then you should describe it clearly and explicitly. And focus on the data analysis itself.

Indeed, we did the complete chain from selection of sample collecting places, DNA isolation, indexing just to the data analysis. The NGS itself was performed externally. In addition, external software and databases (Silva, Galaxy) had been used for processing and quality check of the sequence data as described in the experimental part.

In line 89 you refer to a table 1, which you have removed from the text.

Table 1 is added now.